# Impulsivity, Emotional Dysregulation and Executive Function Deficits Could Be Associated with Alcohol and Drug Abuse in Eating Disorders

**DOI:** 10.3390/jcm9061936

**Published:** 2020-06-21

**Authors:** María Lozano-Madrid, Danielle Clark Bryan, Roser Granero, Isabel Sánchez, Nadine Riesco, Núria Mallorquí-Bagué, Susana Jiménez-Murcia, Janet Treasure, Fernando Fernández-Aranda

**Affiliations:** 1Department of Psychiatry, Bellvitge University Hospital-IDIBELL, 08907 L’Hospitalet de Llobregat, Barcelona, Spain; maria.lozano@bellvitgehospital.cat (M.L.-M.); isasanchez@bellvitgehospital.cat (I.S.); nriesco@bellvitgehospital.cat (N.R.); sjimenez@bellvitgehospital.cat (S.J.-M.); 2Psychiatry and Mental Health Group, Neuroscience Program, Institut d’Investigació Biomèdica de Bellvitge—IDIBELL, 08907 L’Hospitalet de Llobregat, Barcelona, Spain; 3Ciber Fisiopatologia Obesidad y Nutrición (CIBERobn), Instituto Salud Carlos III, 28029 Madrid, Spain; Roser.Granero@uab.cat (R.G.); nmallorqui@live.com (N.M.-B.); 4Department of Psychological Medicine, Section of Eating Disorders, Institute of Psychiatry, Psychology and Neuroscience, King’s College London, SE5 8AF London, UK; danielle.clarkbryan@kcl.ac.uk (D.C.B.); janet.treasure@kcl.ac.uk (J.T.); 5Departament de Psicobiologia i Metodologia. Universitat Autònoma de Barcelona, 08035 Barcelona, Spain; 6Department of Psychiatry, Addictive Behavior Unit, Hospital de la Santa Creu i Sant Pau, Biomedical Research Institute Sant Pau, 08001 Barcelona, Spain; 7Department of Clinical Sciences, School of Medicine and Health Sciences, University of Barcelona, 08907 L’Hospitalet de Llobregat, Barcelona, Spain

**Keywords:** eating disorder, alcohol and/or drug abuse, substance use disorder, executive functions, impulsivity, emotional dysregulation

## Abstract

Background: Empirical data suggests a high comorbid occurrence of eating disorders (EDs) and substance use disorders (SUDs), as well as neurological and psychological shared characteristics. However, no prior study has identified the neuropsychological features of this subgroup. This study examines the prevalence of alcohol and/or drug abuse (A/DA) symptoms in ED patients. It also compares the clinical features and neuropsychological performance of ED patients with and without A/DA symptoms. Methods: 145 participants (74.5% females) with various forms of diagnosed EDs underwent a comprehensive clinical (TCI-R, SCL-90-R and EDI-2) and neuropsychological assessment (Stroop, WCST and IGT). Results: Approximately 19% of ED patients (across ED subtypes) had A/DA symptoms. Those with A/DA symptoms showed more impulsive behaviours and higher levels of interoceptive awareness (EDI-2), somatisation (SCL-90-R) and novelty seeking (TCI-R). This group also had a lower score in the Stroop-words measure, made more perseverative errors in the WCST and showed a weaker learning trajectory in the IGT. Conclusions: ED patients with A/DA symptoms display a specific phenotype characterised by greater impulsive personality, emotional dysregulation and problems with executive control. Patients with these temperamental traits may be at high risk of developing a SUD.

## 1. Introduction

Eating disorders (EDs) are mental illnesses characterized by abnormal eating behaviours, which normally lead to significant impairments in physical health and psychosocial functioning [1]. EDs often co-occur with other untreated mental conditions such as substance use disorders (SUDs) [2], summarised as the inability to control the usage or craving of substances despite problems related to their use [1]. A recent meta-analysis of 43 studies estimated that the lifetime prevalence of any SUD among adults with EDs is 25.4%, with tobacco, caffeine and alcohol most commonly abused [2]. The prevalence of substance abuse varies according to demographic factors (e.g., age, sex, ethnicity) and ED diagnostic subtype [2,3,4]. According to the above-mentioned meta-analysis [2], the lifetime prevalence of SUDs seems to be higher in female-sample studies (26%) compared to mixed studies (15%), in adult populations (26%) compared to young adults (19%) and in primarily Caucasian samples (24%) compared to Asian samples (7%). Similarly, several studies have reported a stronger association between substance abuse and bulimia nervosa (BN) compared to anorexia nervosa (AN) [3,4]. Overall, it seems that those ED patients with purging behaviours are at higher risk of developing a comorbid SUD [2,5].

This comorbidity might be due to shared biological and psychological endophenotypes [6]. Genetic studies reveal that EDs and SUDs share overlapping genetic risk factors, especially those ED subtypes characterized by binge-purge behaviours [7,8]. Personality traits such as impulsivity are also shared [9,10,11], as are the elevated rates of psychopathology (especially social anxiety, antisocial behaviour, and cluster B and C personality disorders) and emotional dysregulation [12,13,14]. Neuropsychological impairments are another common feature [15,16,17,18,19,20,21,22,23,24,25,26,27], especially in executive functions (i.e., high-level cognitive processes implicated in the formation of successful goal-directed behaviours [28]). Difficulties in decision-making, cognitive flexibility and inhibitory control have been reported in all ED subtypes [15,16,17,18,19,20], as well as in SUD patients [21,22,23,24,25,26,27]. However, no studies have explored neuropsychological performance in ED patients with substance abuse comorbidity.

Empirical data on the comorbidity of EDs and SUDs, as well as their biological and psychological shared characteristics are available. However, there are no prior studies which have assessed the neuropsychological profile in ED patients with substance abuse symptomatology. Identifying these neuropsychological features might help to develop specific treatments that target these deficits with the purpose of preventing the later evolution into a SUD. As such, the first aim of this study was to examine the prevalence of alcohol and/or drug abuse (A/DA) symptoms in a heterogeneous series of ED patients. Our hypothesis was that A/DA symptoms would be most common in those with binge-purge behaviours. A second aim was to compare clinical (i.e., eating symptomatology, general psychopathology and personality traits) and neuropsychological features (i.e., decision-making, cognitive flexibility and inhibitory control) of patients with or without A/DA symptoms. Our hypothesis was that ED patients with A/DA symptoms will display higher impulsivity and emotional dysregulation, along with poorer executive functions.

## 2. Materials and Methods

### 2.1. Participants

The total sample comprised of 145 participants (74.5% females) aged between 18 and 60 years old. All participants were diagnosed with an ED by experienced clinicians according to the DSM-5 diagnostic criteria [1]. The final sample included: AN-restrictive subtype [AN-R] (*n* = 57), AN-binge/purge subtype [AN-BP] (*n* = 26), BN (*n* = 28), binge eating disorder [BED] (*n* = 22) and other specified feeding or eating disorder [OSFED] (*n* = 12). 

Participants were recruited at the *Eating Disorders Unit* within the Department of Psychiatry at Bellvitge University Hospital (Barcelona, Spain), where they were receiving outpatient treatment. All participants were informed about the research procedures and gave their informed consent in writing. Procedures were approved by the Ethical Committee of the above-mentioned institution in accordance with the Helsinki Declaration of 1975 as revised in 1983 (reference: PR146/14). Exclusion criteria were the following: (1) history of chronic medical illness or neurological condition that might affect cognitive function; (2) head trauma with loss of consciousness for more than 2 min, learning disability or intellectual disability; (3) use of psycho-active medications or drugs; (4) age under 18 or over 60 (to discard neuropsychological deficits associated with the age).

### 2.2. Procedure and Assessment

As part of the protocol, all patients who arrived at the *Eating Disorders Unit* seeking treatment for an ED were assessed by experienced clinicians using a semi-structured clinical interview based on the DSM-5 diagnostic criteria [1]. All patients consecutively diagnosed with an ED were screened for the inclusion criteria of the study. Those who met the criteria and voluntarily accepted to be part of the study underwent a comprehensive neuropsychological and clinical assessment within the first week of their outpatient treatment. Weight and Body Mass Index (BMI) were measured for all subjects on the day of assessment. Additional sociodemographic information was also taken. The neuropsychological tests were selected to cover various aspects of executive functions including decision-making, response inhibition, strategic planning and cognitive flexibility and were administered by a trained psychologist in a single session and in a randomised order. Finally, information regarding the presence or absence of impulsive behaviours (including alcohol abuse, drugs abuse, binge episodes, theft, kleptomania and compulsive buying) was taken from the semi-structured clinical interview. The presence of A/DA symptoms was defined as the confirmation of current or lifetime behaviours of alcohol abuse, illicit drug abuse or both, causing significant distress or impairments in daily functioning. 

#### 2.2.1. Psychopathological/Personality Measures

*Semi-structured clinical interview*: This interview is based on the ED module of the Structured Clinical Interview for DSM-5 (SCID-5 [29]) and is used to ascertain the presence of a current ED according to the DSM-5 criteria. It provides specific information regarding the symptomatology and course of the ED. It also includes other questions in relation to the impulsive behaviours frequently found in ED patients, such us alcohol abuse, drug abuse, theft, compulsive buying, etc.

The *Temperament and Character Inventory-Revised* (TCI-R [30]; Spanish validation [31]) is a reliable and valid 240-item questionnaire that measures seven personality dimensions: four temperament dimensions (novelty seeking, harm avoidance, reward dependence and persistence) and three about character (self-directedness, cooperativeness and self-transcendence). The scales in the revised version showed a mean internal consistency of 0.87 (*α* coefficient).

The *Symptom Checklist-90 Revised* (SCL-90-R [32]; Spanish validation [33]) is a 90-item questionnaire which evaluates psychopathological symptoms; these are grouped as follows: somatization, obsessive-compulsive, interpersonal sensitive, depression, anxiety, hostility, phobic anxiety, paranoia and psychotic. It also includes a global severity index (GSI), designed to measure overall psychological distress. This instrument has demonstrated satisfactory psychometric properties in the Spanish version, obtaining a mean internal consistency of 0.75 (*α* coefficient).

The *Eating Disorders Inventory-2* (EDI-2 [34]; Spanish validation [35]) is a 91-item self-report questionnaire that assesses the following ED factors: drive for thinness, bulimia, body dissatisfaction, ineffectiveness, perfectionism, interpersonal distrust, interoceptive awareness, maturity fears, asceticism, impulse regulation and social insecurity. This instrument was validated in a Spanish population with a mean internal consistency of 0.63 (*α* coefficient).

#### 2.2.2. Neuropsychological Measures

The Stroop Colour and Word Test (SCWT [36]; Spanish version [37]) is an extensively used neuropsychological test to assess inhibitory control (including response inhibition and interference control). It consists of three different lists: a word list containing the names of colours printed in black ink, a colour list that comprises letter Xs printed in colour and a colour-word list comprised of names of colours in a colour ink that does not match the written name. Three final scores are obtained based on the number of items that the participant is able to read on each of the three lists in a time window of 45 s. An interference score is computed from all three lists. A higher score is interpreted as better inhibitory control.

The Wisconsin Card Sorting Test (WCST [38]) is a computerized set-shifting task for assessing cognitive flexibility. It includes 128 cards that vary according to three attributes: number (N), colour (C) and shape (S). The participant has to pile the cards beneath four reference cards that also vary along these same dimensions, and in order to succeed, they have to settle upon a predetermined sorting rule. The only feedback given to the participant is the word “right” or “wrong” after each sorting. Initially, C is the correct sorting category, and positive feedback is given only if the card is placed in the pile with the same colour. After 10 consecutive correct sorts, the rule changes. Thus, the positive feedback is only given when the sorting matches the new category. By trial and error, the participant must learn to change the sorting categories according to the given feedback. There are up to six attempts to derive a rule, providing rule shifts in the following category sequence: C-S-N-C-S-N. Participants are not informed of the correct sorting principle and that the sorting principle shifts during the test. The test is completed when all 128 cards are sorted or after the six full categories are completed. The number of completed categories, the percentage of perseverative errors (i.e., failures to change sorting strategy after negative feedback) and the percentage of non-perseverative errors are recorded.

The Iowa Gambling Task (IGT [39]) is a computerized task to evaluate decision-making, which has also been proposed as a measure of choice impulsivity [40]. It involves a total of 100 turns distributed across four decks of cards (A, B, C and D), and each time the participant selects a deck, a specified amount of play money is awarded. The interspersed rewards among these decks are probabilistic punishments (monetary losses with different amounts). Participants are instructed that the final aim of the task is to win as much money as possible and to avoid losing as much money as possible. Moreover, they may choose cards from any deck, and switch decks at any time. This test is scored by subtracting the number of cards selected from decks A and B from the number of cards selected from decks C and D. Decks A and B are not advantageous as the final loss is higher than the final gain; however, decks C and D are advantageous since the punishments are smaller. Higher scores indicate better performance on the task.

### 2.3. Statistical Analyses

Statistical analysis was carried out with Stata16 for Windows [41]. The comparisons between the groups with and without A/DA symptoms were based on T-TEST procedures for quantitative variables and chi-square tests (χ^2^) for categorical variables. The effect size of the mean differences in the clinical variables was estimated with Cohen’s-*d* coefficient, considering null effect for |*d*| < 0.20, low-poor for |*d*| > 0.20, moderate-medium for |*d*| > 0.5 and large-high for |*d*| > 0.8 [42]. The significance tests were complemented with other standardised measures of the effect size: partial eta-squared coefficients (*η*^2^), which measures the proportion of the total variance in a criterion associated with the membership of the different groups, defined by the independent variable once the potential effects of other predictors and interactions are partialled out (in one way ANOVA, eta-squared and partial eta-squared come out the same, but in multivariate ANOVA, their values differ). The comparisons between the proportions were based on the Cohen’s-*h* coefficient, a standardised measure of the distance between the proportions obtained in two groups; it is estimated as the difference of the arcsine transformation for the two probabilities [43]. In addition, and due the multiple statistical comparisons performed on the clinical and neuropsychological variables, Finner-correction was used to control the increase in the Type-I error [44]. Finner-method uses a stepwise multiple comparisons procedure, which solves the monotonicity of the critical values by means of an inequality for the distribution function of the statistic range, using the principle of family-wise Type I correction. The post-hoc power calculation was also conducted for each observed effect based on the sample size and the parameter estimates, defining an alpha value *α* = 0.05.

A 2 × 5 mixed analysis of variance was obtained to analyse the learning curve in the IGT test. For this analysis, the group was defined as the between-subjects factor (2 levels: with versus without A/DA) and the IGT-block as the within-subjects factor (5 levels: blocks 1 to 5). Tests of within-subjects included polynomial contrasts for assessing the presence of trends in the mean estimates (linear, quadratic, cubic and order-4). Effect size of the parameter estimates were assessed with eta-squared values (*η*^2^). 

## 3. Results

### 3.1. Characteristics of the Sample

Most participants were women (108, 74.5%), born in Spain (130, 89.7%) and single (105, 72.4%). Mean chronological age was 30.3 years (SD = 10.3), age of ED onset was 22.7 years (SD = 9.1) and duration of the ED symptoms 7.7 years (SD = 7.4). The prevalence of impulsive behaviours was 43.1% for binges episodes, 18.6% for theft, 3.4% for kleptomania and 11.0% for compulsive buying. No participants reported instances of problematic gambling behaviours.

Table 1 shows the comparison between the groups with and without A/DA symptoms. Statistical differences were found for the presence of impulsive behaviours (higher prevalence among patients with A/DA).

### 3.2. Prevalence of A/DA in ED Patients

The number of patients with A/DA symptoms was 27 (prevalence = 18.6%). Table 2 compares prevalence estimates between ED subtypes and gender. No differences by gender (women (18.5%) and men (18.9%) (χ^2^ = 0.01, *df* = 1, *p* = 0.957)) or subtype (14.0% for AN-R, 23.1% for AN-BP, 25.0% for BN, 18.2% for BED and 16.7% for OSFED (χ^2^ = 1.92, *df* = 4, *p* = 0.751)) were found. The highest effect size for the comparison between AN-R and BN was a Cohen’s-h into the low range |*h*| = 0.28.

### 3.3. Comparison of the Clinical and Neuropsychological Profile of Patients with and without A/DA 

Table 3 compares the clinical measures of patients with and without A/DA symptoms. Those with A/DA had higher levels of interoceptive awareness (EDI-2), somatisation (SCL-90-R) and novelty seeking (TCI-R). Internal consistency in the sample of the study was between good to excellent for all the psychometrical scales (Cronbach’s alpha values).

Table 4 shows the comparison of the neuropsychological profile. Higher scores in perseverative errors (WCST) and in the first block of the IGT and lower scores in the Stroop-words measure were found.

The results of the mixed ANOVA obtained to analyse the learning curve in the IGT showed a quasi-significant interaction parameter IGT-by-Group (*F*_(3.42;488.5)_ = 2.02, *p* = 0.091, *η*^2^ = 0.014; Greenhouse–Geisser adjusted) as well as a main effect of the IGT-block (*F*_(3.42;488.5)_ = 2.65, *p* = 0.041, *η*^2^ = 0.018). Figure 1 shows the adjusted mean net scores in the blocks, showing a learning trajectory only for patients without A/DA symptoms: Among this group, polynomial contrasts for the IGT-block showed a significant linear trend (*F*_(1;117)_ = 18.3, *p <* 0.001, *η*^2^ = 0.135) and a significant quadratic trend (*F*_(1;117)_ = 6.97, *p* = 0.009, *η*^2^ = 0.056), while non-significant results were found for the cubic (*F*_(1;117)_ = 0.01, *p* = 0.931 *η*^2^ = 0.001) and order 4 (*F*_(1;117)_ = 0.59, *p* = 0.442, *η*^2^ = 0.005) trends. In patients without A/DA symptoms, post-hoc pairwise comparisons defining the difference-type contrasts (based on comparing each block with the previous) showed statistical differences between blocks 2 versus block 1 (*p* = 0.008) and block 3 versus block 2 (*p* = 0.011), while no statistical differences were found between blocks 4 versus block 3 (*p* = 0.886) and block 5 versus block 4 (*p* = 0.832). However, patients with A/DA symptoms did not have significant results (linear: *F*_(1;26)_ = 0.27, *p* = 0.607, *η*^2^ = 0.010; quadratic: *F*_(1;26)_ = 0.06, *p* = 0.815, *η*^2^ = 0.002; cubic: *F*_(1;26)_ = 0.01, *p* = 0.988, *η*^2^ = 0.001; order 4: *F*_(1;26)_ = 2.30, *p* = 0.141, *η*^2^ = 0.081). No statistical differences were found in the post-hoc pairwise comparisons comparing the IGT blocks among patients who reported the presence of A/DA.

## 4. Discussion

The first aim of this study was to assess the prevalence of A/DA symptoms in a sample of patients across the range of ED diagnoses. The lifetime A/DA prevalence was similar to that found previously in ED patients [2], although slightly lower. This small difference could be explained by the fact that we only assessed symptoms of alcohol and illicit drugs abuse, whereas previous studies considered other substances such as tobacco and caffeine [2]. Although this study had insufficient power to detect differences between subtypes, the trends were in the same direction as previous studies, with BN patients showing the highest prevalence [2,3,4,5] and those with AN-BP higher than those with AN-R [45], confirming our first hypothesis based on previous studies. We failed to find the gender difference in prevalence observed in a previous systematic review [2]; however, this might be a question of low power and methodology. While our study compares female and male patients within the same sample, Bahji et al. [2] compared the prevalence observed in female sample studies to mixed studies. 

When comparing the clinical profile, the A/DA group scored higher on interoceptive awareness subscale (EDI-2), which indicates a poorer ability to recognise and differentiate between hunger and satiety and emotional states [34]. In ED patients, emotional dysregulation is associated with difficulties in controlling impulsive behaviours (e.g., compulsive binging, purging and overeating) in both negative and positive states [46,47], and this may lead to abuse of alcohol and drugs [13,14]. Indeed, the weaker interoceptive awareness previously found in individuals with BN [48] may explain their higher vulnerability to develop comorbid A/DA symptoms [3,4].

Moreover, the A/DA group had higher levels of somatisation (SCL-90-R), the experience of psychological distress in the presence of unexplained physical symptoms [49]. It is suggested that patients experiencing high levels of somatising symptoms use substances to numb uncomfortable experiences [49]. Furthermore, somatising symptoms are closely associated with symptoms of substance intoxication or withdrawal, but whether they are a cause or effect of substance abuse remains unclear [50]. In any case, our results seem to reflect this association between somatising and substance abuse symptoms, yet longitudinal studies would be needed to clarify the direction of these interactions.

The group with A/DA symptoms had higher scores for the novelty seeking trait (TCI-R), mainly related to impulsivity [30]. Our findings align with other neurobehavioural research, which suggests that novelty seeking is associated with vulnerability to substance abuse [10,11] and with the finding of a higher prevalence of binge episodes, theft, kleptomania and compulsive buying in this subgroup. Strong associations among all the impulsive behaviours mentioned above are documented in the literature. For instance, ED patients who engage in binge-purge behaviours frequently display high prevalence of SUD [2,3,4,5,45] and compulsive buying [51,52]. Similarly, several studies have reported high rates of SUDs in individuals with kleptomania and compulsive buying [53,54]. 

The neuropsychological assessment results revealed that ED patients with A/DA symptoms display significantly lower scores in the Stroop-words measure, demonstrating a poorer reading ability in terms of speed and accuracy [36]. Contrary to expectations, no differences between groups were found in the Stroop-interference measure, meaning no differences in inhibitory control [36]. Impairments in inhibitory control are found to be a core deficit among binge/purge-type EDs [15,16] and SUDs [21,23], especially when examining stimuli related to each disorder (e.g., food, body shape, use of substances, etc.) [15,55,56]. Since inhibitory deficits seem to represent a risk factor for the development of both disorders separately, we expected to find greater inhibitory control impairments when both disorders co-occur. The fact that we failed to find this result might be explained because of the limited statistical power of this study. Further studies with larger samples are needed to clarify this issue.

In addition, patients with A/DA symptoms made more perseverative errors (WCST). Previous research demonstrated that individuals with EDs perform worse on set-shifting tasks than healthy individuals, which translates to poorer cognitive flexibility and higher rigidity [17,18]. Reduced cognitive flexibility has also been observed in individuals with SUDs, which may negatively impact on their problem-solving strategies [24,25]. In summary, poor cognitive flexibility is shown to be a common feature in EDs and SUDs, and our results point out that it might be highlighted when both disorders are present at the same time. However, due to the relatively small sample size of one of the groups and the low statistical power, our results should be interpreted with caution until more evidence is available. 

Lastly, even though no differences between groups were found on the IGT total score, both groups displayed an impaired decision-making performance in comparison to healthy population [18,57,58]. However, when comparing the learning curve of both groups, patients with A/DA symptoms showed a weaker learning trajectory and greater difficulties to learn the reward/punishment contingencies of their choices [59]. Our results point out that decision-making problems seem to be present in ED patients [18,19,20], particularly in those with comorbid substance abuse symptomatology. Once again, future studies are needed to confirm this finding. 

The results of this study should be interpreted in the context of some limitations. First, although our sample size is relatively large, the subgroups vary in size, and one of them is relatively small, leading to a low statistical power. Studies with small samples are severely underpowered, which might represent a concern if the “null hypothesis” cannot be rejected. For this reason, the significance tests were complemented with other standardized measure of the effect size. In any case, the results of this study should be interpreted with caution and considering this limitation. Furthermore, because the design is cross-sectional, claims regarding causality cannot be made. Future longitudinal studies should examine the extent to which psychopathology and cognitive function improve after treatment. Another limitation was the lack of a formal diagnosis of SUD in our sample. Finally, our sample was largely made up of young adults, and it would be of clinical interest to explore whether similar impairments in neuropsychological functioning are present in older samples. 

## 5. Conclusions

To conclude, although previous studies have explored comorbidity between EDs and SUDs, this is the first study to explore gender differences. Moreover, it is the first to assess the neuropsychological profile in ED patients with A/DA symptoms. We found that this subgroup displays a specific phenotype characterised by greater impulsivity (i.e., high novelty seeking and difficulties to control impulsive behaviours), noticeable emotional dysregulation (i.e., decreased interoceptive awareness) and more impaired executive control (i.e., low cognitive flexibility and poor decision-making). Of future benefit would be the consolidation of our findings in larger samples and clarifying if these deficits are involved in the development and maintenance of substance abuse comorbidity. As such, this subgroup of ED patients might benefit from augmented treatment that targets these problems, such as inhibitory control training [60], emotion regulation training [61] or cognitive remediation therapy [62]. This may reduce the present substance abuse symptomatology and prevent later evolution into a SUD, as previous studies have reported that ED patients who do not receive any adjunctive treatment for substance abuse are at high risk for switching from one problematic behaviour to the other, especially during the recovery process [2].

## Figures and Tables

**Figure 1 jcm-09-01936-f001:**
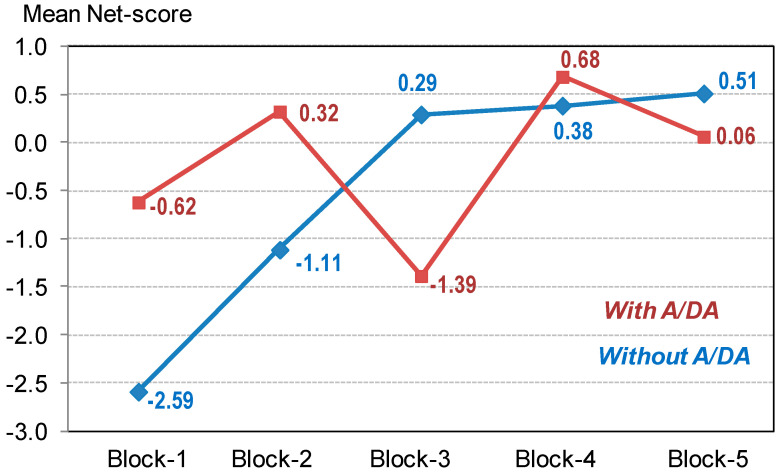
Learning curve in the IGT measure. Note. A/DA: alcohol and/or drug abuse. Sample size: *n* = 145.

**Table 1 jcm-09-01936-t001:** Descriptive of the sample.

		Without A/DA		With A/DA		*p*
		*n* = 118		*n* = 27		
		*n*	%	*n*	%	
Sex	Women	88	74.6%	20	74.1%	0.957
	Men	30	25.4%	7	25.9%	
ED subtype	Anorexia restrictive	49	41.5%	8	29.6%	0.751
	Anorexia binge/purge	20	16.9%	6	22.2%	
	Bulimia	21	17.8%	7	25.9%	
	Binge eating disorder	18	15.3%	4	14.8%	
	Other specified feeding eating dis.	10	8.5%	2	7.4%	
Education	Primary	30	25.4%	10	37.0%	0.438
	Secondary	52	44.1%	11	40.7%	
	University	36	30.5%	6	22.2%	
		Mean	SD	Mean	SD	*p*
Age (years-old)		30.30	10.13	30.56	11.22	0.907
Onset of the ED (years-old)		22.94	8.99	21.59	9.83	0.491
Duration of the ED (years)		7.36	7.58	8.96	6.72	0.312
BMI (kg/m^2^)		21.70	8.47	22.50	8.87	0.663
Other impulsive behaviours		*n*	%	*n*	%	*p*
Binges episodes		46	39.3%	16	59.3%	0.049 *
Theft		18	15.3%	9	33.3%	0.029 *
Kleptomania		2	1.7%	3	11.1%	0.045 *
Compulsive buying		10	8.5%	6	22.2%	0.040 *

A/DA: alcohol and/or drugs abuse symptoms. ED: eating disorder. BMI: body mass index. SD: standard deviation. *: significant comparison (0.05 level).

**Table 2 jcm-09-01936-t002:** Prevalence estimate of A/DA depending on ED subtype and gender.

Total		AN-R		AN-BP		BN		BED		OSFED		Women		Men	
*n* = 145		*n* = 57		*n* = 26		*n* = 28		*n* = 22		*n* = 12		*n* = 108		*n* = 37	
*n*	%	*n*	%	*n*	%	*n*	%	*n*	%	*n*	%	*n*	%	*n*	%
27	18.6%	8	14.0%	6	23.1%	7	25.0%	4	18.2%	2	16.7%	20	18.5%	7	18.9%
	χ^2^	1.92	(*df* = 4)									0.01	(*df* = 1)		
	*p*	0.751										0.957			

AN-R: anorexia restrictive. AN-BP: anorexia binge/purge. BN: bulimia. BED: binge eating disorder. OSFED: other specified feeding eating disorder. df: degrees of freedom.

**Table 3 jcm-09-01936-t003:** Comparison of the clinical profile between groups with and without A/DA.

	*α*	Without A/DA		With A/DA		*p*	*|d|*	*η* ^2^	Power
		*n* = 118		*n* = 27					
		Mean	SD	Mean	SD				
EDI Drive for thinness	0.846	11.24	6.65	12.10	6.50	0.544	0.13	0.003	0.093
EDI Body dissatisfaction	0.922	13.35	8.86	15.66	7.70	0.213	0.28	0.011	0.237
EDI Interoceptive awareness	0.868	8.98	7.13	12.10	5.95	0.036 *	0.52 ^†^	0.030	0.556
EDI Bulimia	0.791	4.92	5.20	6.51	4.83	0.148	0.32	0.015	0.303
EDI Interpersonal distrust	0.824	5.59	4.84	5.92	4.47	0.752	0.07	0.001	0.061
EDI Ineffectiveness	0.905	8.61	6.89	10.78	6.91	0.142	0.32	0.015	0.312
EDI Maturity fears	0.841	7.54	5.08	7.58	5.89	0.975	0.01	0.001	0.050
EDI Perfectionism	0.842	5.31	4.40	5.16	4.54	0.878	0.03	0.001	0.053
EDI Impulse regulation	0.850	5.43	5.85	5.60	4.43	0.886	0.03	0.001	0.052
EDI Ascetic	0.703	5.99	4.37	6.92	3.76	0.307	0.23	0.007	0.175
EDI Social insecurity	0.825	6.37	5.17	7.28	5.72	0.417	0.17	0.005	0.127
EDI Total score	0.974	83.33	48.73	95.62	42.19	0.228	0.27	0.010	0.225
SCL-90R Somatization	0.857	1.45	0.90	1.82	0.77	0.049 *	0.44	0.026	0.497
SCL-90-R Obsessive-compulsive	0.909	1.57	0.98	1.78	0.89	0.308	0.22	0.007	0.174
SCL-90-R Interpersonal sensitive	0.912	1.76	1.04	1.94	0.91	0.416	0.18	0.005	0.128
SCL-90-R Depression	0.946	2.00	1.05	2.21	0.82	0.329	0.22	0.007	0.164
SCL-90-R Anxiety	0.911	1.38	0.95	1.66	0.74	0.145	0.34	0.015	0.307
SCL-90-R Hostility	0.774	1.17	0.88	1.25	0.97	0.672	0.09	0.001	0.071
SCL-90-R Phobic anxiety	0.854	0.78	0.87	0.86	0.73	0.667	0.10	0.001	0.072
SCL-90-R Paranoia	0.875	1.27	0.89	1.32	0.81	0.821	0.05	0.001	0.056
SCL-90-R Psychotic	0.893	1.17	0.77	1.28	0.67	0.529	0.14	0.003	0.096
SCL-90-R GSI	0.980	1.50	0.82	1.69	0.68	0.257	0.26	0.009	0.204
SCL-90-R PST	0.980	56.61	19.91	63.15	17.64	0.119	0.35	0.017	0.345
SCL-90-R PSDI	0.980	2.20	0.66	2.32	0.42	0.383	0.21	0.005	0.140
TCI-R Novelty seeking	0.797	94.68	13.80	103.37	13.83	0.004 *	0.63 ^†^	0.057	0.834
TCI-R Harm avoidance	0.925	112.31	20.84	116.07	21.50	0.402	0.18	0.005	0.133
TCI-R Reward dependence	0.700	101.21	15.47	102.56	17.33	0.691	0.08	0.001	0.068
TCI-R Persistence	0.860	114.12	20.30	107.33	24.13	0.133	0.30	0.016	0.323
TCI-R Self-directedness	0.908	125.72	21.84	119.33	24.19	0.181	0.28	0.012	0.266
TCI-R Cooperativeness	0.831	136.66	17.28	135.89	15.97	0.832	0.05	0.001	0.055
TCI-R Self-transcendence	0.893	63.16	15.28	63.37	18.91	0.951	0.01	0.001	0.050

SD: standard deviation. Cronbach’s alpha in the sample. *: significant comparison (0.05 level). ^†^: effect size into the moderate (*|d|* > 0.50) to high range (*|d|* > 0.80). *p*-values with Finner-correction.

**Table 4 jcm-09-01936-t004:** Comparison of the neuropsychological profile between groups with and without A/DA.

	Without A/DA		With A/DA		*p*	*|d|*	*η* ^2^	Power
	*n* = 118		*n* = 27					
	Mean	SD	Mean	SD				
Stroop Words	104.43	19.39	95.61	24.53	0.045 *	0.40	0.028	0.521
Stroop Colour	75.84	16.06	80.24	19.74	0.221	0.24	0.010	0.231
Stroop Words-colour	48.27	11.75	46.50	9.33	0.467	0.17	0.004	0.112
Stroop Interference	5.09	8.99	4.42	7.92	0.725	0.08	0.001	0.064
WCST Total trials	94.42	20.72	100.55	20.70	0.168	0.30	0.013	0.280
WCST Correct	67.83	10.87	68.06	14.02	0.924	0.02	0.001	0.051
WCST Perseverative errors	12.40	10.33	18.54	19.41	0.023 *	0.39	0.036	0.629
WCST Non-perseverative errors	14.20	15.25	14.22	11.91	0.993	0.01	0.001	0.050
WCST Conceptual	60.64	16.54	59.18	19.97	0.692	0.08	0.001	0.068
WCST Categories completed	5.10	1.80	4.88	1.98	0.577	0.12	0.002	0.086
WCST Trials completed 1st categ.	20.36	24.17	26.73	31.78	0.248	0.23	0.009	0.210
IGT Block 1	−2.59	4.21	−0.62	2.84	0.023 *	0.55 ^†^	0.036	0.629
IGT Block 2	−1.11	4.91	0.32	4.41	0.165	0.31	0.013	0.284
IGT Block 3	0.29	5.80	−1.39	3.55	0.152	0.35	0.014	0.298
IGT Block 4	0.38	6.69	0.68	6.67	0.834	0.04	0.001	0.055
IGT Block 5	0.51	7.78	0.06	5.42	0.777	0.07	0.001	0.059
IGT Total	−2.52	20.79	−0.95	11.51	0.704	0.09	0.001	0.067

SD: standard deviation. *: significant comparison (0.05 level). ^†^: effect size into the moderate (*|d|* > 0.50) to high range (*|d|* > 0.80). *p*-values with Finner-correction.

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
