# Peer review of "Impulsivity, Emotional Dysregulation and Executive Function Deficits Could Be Associated with Alcohol and Drug Abuse in Eating Disorders"

_jcm, 2020, doi:10.3390/jcm9061936_

Round 1
Reviewer 1 Report
This paper presents novel findings comparing ED patients to ED/Substance abuse patients. There is not a lot of research in this area, although we may clinically have ideas about how these patients compare to each other. Thus, it is nice to have more research in this area as opposed to just relying on clinical observation.
The paper is clearly written, and I don't think it requires much editing.
One thing I was wondering about was exactly what kind of patients these were. The article states that they were recruited from an ED unit in a hospital. Were they all inpatients or were some at different levels of care? Based on their average BMI, the AN patients do not appear particularly underweight. Were these patients nearing the end of their treatment episode or were they already in recovery?
In lines 254-261, authors discuss how poor interoceptive awareness causes A/DA but then make the point that it's not clear whether somaticizing is a cause or effect of substance use. Seems that the same point could and should be made for both interoceptive awareness and somaticizing.
In lines 270-277, it is very counterintuitive that you would not see differences in inhibitory control in these two groups. This might bear more explanation and I'm curious if it matches up with past research findings (even though this particular measure may not have been used, I think other researchers have looked at inhibition in these groups).
Author Response
"Please see the attachment."

Reviewer 2 Report
The paper “Could impulsivity, emotional dysregulation and executive function deficits increase the risk of developing alcohol and drug abuse in eating disorders?” reports on a study that investigated the prevalence of alcohol and drug abuse in eating disorders. Furthermore the study examined within a group of patients with eating disorders the difference between patients who also have alcohol and drug abuse (A/DA) symptoms and those who don’t have these symptoms in temperament and character (TCI), in psychopathological symptoms (SCL-90), in eating disorder related psychopathology (EDI) and in several executive functions (Stroop Colour and Word Test, Wisconsin Card Sorting Test and Iowa Gambling Task). The idea behind the study is nice. The goal of the study is clear and the study is straightforward. However, I do have several remarks and questions for the authors.
My main concern is the sample size, 145 participants seems quite substantial. However since only 27 persons have A/DA symptoms, and very many variables (31 self-report scales [12 EDI, 12 SCL-90 and 7 TCI] and 17 executive function variables [4 Stroop, 7 WCST, and 6 IGT]) are being compared between the two groups, it is hardly a surprise that only a few “significant” results were found. And as a consequence, it is not clear how these results should be interpreted. A statistical power calculation is missing, and should be added to the paper. I performed a post-hoc power calculation based on the sample size and effect sizes reported in this study, and this shows that there was only adequate statistical power to detect larger effect sizes (effect size≥ 0.6; β ≥ 0.88).
Because of this limited sample size, the comparison of the presence of A/DA symptoms between the five ED subtypes, is completely underpowered. And the added value of this analysis is questionable.
The other main issue is the fact that so many variables were being compared. The authors mention in the Methods-Statistical Analyses section (page 5 line 173-174) that a Finner-correction was used to control the increase in type-I error. Are the p-values that are presented in Table 3 (page 7) and Table 4 (page 8) corrected with this approach? It is not clear, because the Finner-correction or the consequences of applying this correction were not mentioned anywhere in the Result section. A Bonferroni correction is a very strict way to correct for multiple testing, but if I look at the number of different T-tests that were performed (48!), I do not think that any of the found differences would still be significant after Bonferroni correction. I do not say that the authors need to perform this approach, but it is necessary to correct for multiple testing and show what happens if you do. In general, I think the number of comparisons is too much, especially in this small sample size. I would recommend to perform some kind of data reduction. Or use more sophisticated statistical modelling to overcome the multiple testing issue.
Another major question is how were the A/DA symptoms assessed? What measurement instrument was used, and what was the definition of having alcohol or drug abuse. Since this is the main subject of the paper, it was surprising that the definition is not mentioned anywhere in the paper.
Other remarks:
I think the title “Could impulsivity, emotional dysregulation and executive function deficits increase the risk of developing alcohol and drug abuse in eating disorders?” is a bit misleading. Since the authors performed cross-sectional analyses it is impossible to answer this question with this design. You can only say that executive function deficits, impulsivity and emotional dysregulation could be associated with alcohol and drug abuse, but not that it increases the risk of developing alcohol and drug abuse.
The introduction is quite short, more explanation in the introduction is needed: why this study is being performed (e.g. why is it important to examine neuropsychological performance in patients with ED and SUD?), and what we can do with the findings of this study. In addition, some lines are not completely clear to me:
Line 74; I would recommend that if the authors mention that substance abuse varies according to demographic factors (e.g. age, sex, ethnicity), that they also mention how. So who have higher or lower rates of substance abuse young/old, men/women, which ethnicity?
Line 76-77: “It seems that those with purging behaviours are at higher risk for comorbidity” I do not completely understand what the authors mean with this sentence. Are these individuals at higher risk for all comorbidity? And what does this have to do with substance abuse?
Line 77-79: “Other forms of shared comorbidity….” What do the authors mean? Are these forms of shared comorbidity between ED and SUDs? In that case, “other” does not make sense, since this is the first time it is mentioned.
In the description of the SCL-90 (page 4 line 124-132) the different scales are not mentioned, but they are used in the analyses. So I would mention them in de Method section as well.
In the second line of the Results section, page 5 line 184, the authors mention the prevalence of “other” impulsive behaviours. Other than what? In the first line of the Results section no impulsive behaviours are mentioned. It is not clear how the A/ DA symptoms were assessed and defined(as I mentioned earlier), in addition it is also not clear how the impulsive behaviours mentioned in this line were assessed.
The Discussion section is very short. The conclusions that were made by the authors in the fourth (page 10 line 275-277) and fifth (page 10, line 282-284) paragraph are a bit strong. “..our results illustrate that inhibitory control is not particularly impaired when both disorders co-occur.” Because of the smaller sample size and the limited statistical power of this study, it is in my opinion a bit risky to make conclusions based on the fact that no significant differences were found between the two groups. It is possible that these effects would be found in larger study populations, especially when it is contrary to expectations.
“…poor cognitive flexibility is shown to be a common feature in EDs and SUDs, which seems to be highlighted when both disorders are present at the same time” again this conclusion is quite strong, especially since it is based on such a small sample in which so many comparisons were made.
In the limitations (line 292-293) the authors should discuss the limited sample size more extensively. It is a major limitation that has large impact on the study, not just a small issue.
I was really surprised by the final line in the Conclusions (line 307-309). Nowhere in the paper biomarkers are mentioned, and in the final sentence the authors say the identification of possible biomarkers would be of future benefit. I don’t understand what biomarkers have to do with this study, but if the authors want to discuss them in the paper, a better introduction and explanation is needed.
Finally, the authors have used a lot of references (78!) for a study that is not very complicated. I do not think it is necessary to refer to 78 other studies.
Author Response
"Please see the attachment."

Round 2
Reviewer 2 Report
The authors have addressed most of the comments on the original version of the paper. However, after reading the revised version of the manuscript, some comments remained and additional comments came up.
The relatively small subsample size in combination with the large number of T-tests performed is/remains the soft spot of the study. My main concern with this is that if you test so many variables in such a small comparison, how can you interpret the results? How robust, valid and relevant are these findings? I think the authors were very responsive and fast to my initial review, and I commend them for that. They have adjusted the Discussion section, so it is clear that these results should be interpreted with caution. The fact that the authors used comprehensive clinical and neuropsychological assessments, that they are honest about the number of statistical tests that they apply and that they perform a correction for the multiple testing, is all very positive.
I want to suggest some additional minor adjustments to enable the readers to see this study in the right perspective.
First, as I mentioned in my initial review I think the authors should add a statistical power calculation to the paper.
Second, could the authors add a small explanation on the Finner correction in the Methods section. What does this correction do?
Thirdly, it is great that the authors have added the partial eta-squared coefficient to the Methods
and Results, could they also add some explanation on how to interpret these coefficients?
Furthermore, I have a remark on the final part of the last sentence of the paper (page 10 line 356-357): “As such, this subgroup might benefit from augmented treatment that targets these deficits, such as inhibitory control training [59], which may prevent the later evolution into a SUD.”
Based on this cross-sectional study it is impossible to say that the neuropsychological problems that the patients with ED and A/DA symptoms have can cause or evolve into SUD. I don’t know if studies have been performed to investigate which one of the disorders comes first, but I don’t believe that they were referred to in this study. So I think the final part of this sentence is debatable.
Finally, Methods section, page 3 line 106: it should be experienced instead of experience clinicians
